

# The Female Athlete Triad—the impact of running and type of diet on the regularity of the menstrual cycle assessed for recreational runners

Joanna Witkoś[1] and Magdalena Hartman-Petrycka[2]

[1] Faculty of Medicine and Health Science, Andrzej Frycz Modrzewski Krakow University, Kraków, Poland
[2] Department of Basic Biomedical Science, Faculty of Pharmaceutical Sciences in Sosnowiec, Medical University of Silesia, Katowice, Poland

## ABSTRACT

**Background**. The Female Athlete Triad (FAT) included three interrelated conditions including disordered eating, amenorrhea, and osteoporosis. The American College of Sports Medicine updated the definition of FAT to reflect the interdependence of low energy availability with or without eating disorders. The main aim of the study was to assess the impact of recreational running on potential disturbances in the regularity of women's menstrual cycles. Additionally, this work compared differences in the menstrual cycle between women runners and women who did not regularly practice sports. The respondents were also asked about the type of diet they consumed.

**Methods**. A total of 360 women took part in the research. This group included 217 runners and 143 control. The authors' questionnaire was used in the research.

**Results**. When compared to the control group, the runners had an increased frequency of menstrual cycles of <24 days (10.14% *vs.* 3.50%), fewer typical cycles of 25–31 days (75.58% *vs.* 86.71%), had fewer regular cycles per year (9.62 *vs.* 11.22), shorter duration of bleeding (4.79 *vs.* 5.27 days), and an increased frequency of painless menstruation (23.96% *vs.* 7.69%). A positive predictor of menstrual cycle disorders was the use of a 'special diet' (R:1.67; 95% C:0.47–2.87).

**Conclusions**. The runners had shorter and less regular monthly cycles and shorter and more often painless menstrual bleeding when compared to the control group. The frequency of menstrual disorders in runners was increased by following a 'special diet'. The frequency of menstrual cycle disorders in runners, however, did not differ significantly from the control group.

# INTRODUCTION

Many health benefits, both sociological as well as psychological, are associated with the systematic participation in physical activity. An imbalance, however, between the level of energy supplied by the diet and the body's energy expenditure leads to serious health effects (*Mountjoy et al., 2018*).

Corresponding author
Joanna Witkoś, jwitkos@afm.edu.pl

The Female Athlete Triad (FAT) was first defined in 1992 and included three interrelated conditions including disordered eating, amenorrhea, and osteoporosis. In 2007, the American College of Sports Medicine updated the definition of FAT to reflect the interdependence of low energy availability (LEA) (with or without eating disorders) with amenorrhea, decreased bone mineral density, and osteoporosis, which may result in the non-simultaneous occurrence of the clinical symptoms (*Matzkin, Curry & Whitlock, 2015*; *De Souza et al., 2017*; *Beals & Meyer, 2017*; *Daily & Stumbo, 2018*; *Williams, Statuta & Austion, 2017*; *Stefani et al., 2016*; *George, Leonard & Hutchnson, 2011*). Subsequently, in 2014, the International Olympic Committee reported the syndrome referred to as relative energy deficiency in sport (RED-S), which highlighted the impact of low energy availability on the human body and recognized FAT as one of the outcomes of LEA in the RED-S model (*Mountjoy et al., 2018*).

In a situation of high energy expenditure, such as that related to participation in sports, an insufficient energy intake from food results in an energy deficit. This deficit inhibits reproductive functions and results in a state referred to as functional hypothalamic amenorrhea.

Occurrence of the three interrelated physiological effects related to LEA has been repeatedly documented and discussed in the research (*Schofield, Thorpe & Sims, 2020*; *Sim & Burns, 2021*; *Civil et al., 2019*). Currently, however, it has been suggested that the presence of at least one component of the FAT in women who practice sports is sufficient to define FAT, because symptoms of the related physiological impacts occur with a different intensity and may appear at different times depending on the athlete's diet and training load (*Williams, Mallinson & De Souza, 2019*). Research has shown that the prevalence of any one component of FAT ranged from 16% to 60% and any two from 3% to 27%, which shows that, for most female athletes, the individual components of FAT may have different time sequences and not always appear simultaneously (*Matzkin, Curry & Whitlock, 2015*; *De Souza et al., 2017*; *Beals & Meyer, 2017*; *Daily & Stumbo, 2018*; *Williams, Statuta & Austion, 2017*; *Stefani et al., 2016*). The simultaneous incidence of all components of FAT is relatively low and varies. A report that reviewed the literature related to single and combined triad markers reported that the incidence of energy deficiency and disordered eating was 0%–48% and 7.1%–89.2%, respectively (*Williams, Statuta & Austion, 2017*).

Until recently, FAT studies only included women in competitive training since it was believed that only professional athletes were susceptible to developing the triad. It is now known that the occurrence of the FAT syndrome may also affect women who are not elite athletes, and the severity of the syndrome varies depending on the type of sport a woman undertakes. It has been shown that FAT most often affects women who take part in those sports and physical activities that include an aesthetic element that favors a slim figure and low body weight, such as artistic gymnastics, figure skating, and also ballet and dance (*Tosi et al., 2019*). These require both a large energy expenditure and a slim, shapely figure. It has also been shown that high morbidity affects women participating in endurance sports, such as cross-country running, cross-country skiing, and the triathlon (*Castelo-Branco et al., 2006*; *Sawai et al., 2018*; *Lynch & Hoch, 2010*; *Thompson, 2007*).

Due to the fact that women today participate in sport at all levels, it has been suggested that more emphasis should be placed on educating women about the occurrence of the FAT (*Westerbeek, 2009*). Most of the scientific literature on RED-S or FAT focuses on professional athletes and does not check the prevalence of, for example, FAT in women who engage in recreational sports. Running is a physical activity that is generally available to most people and can be practiced as a stand-alone sport or as an integral part of many training programs. Short- and long-distance running are employed as both recreational and professional sports. Women undertake sports activities for many reasons, including typical sports rivalry (competitions, marathons, olympics), but also for recreational purposes, social interactions, health reasons such as weight reduction, and improvement of general physical fitness. Physically active women are generally healthier than people leading a sedentary lifestyle and have greater self-confidence that results from an increase in their self-esteem and physical attractiveness (*Macleod, 1998*). Moreover, sport, especially when undertaken on a recreational basis, prevents depression and anxiety (*Tosi et al., 2019*; *Sawai et al., 2018*;*Lynch & Hoch, 2010*; *Thompson, 2007*). Regardless of the motivation for this type of physical activity, it is important to understand the impact that running can have on the overall health of women who run for recreational purposes.

The main aim of the study was to assess the impact that recreational running has on potential disturbances in the regularity of women's menstrual cycles. Additionally, it evaluated the differences in the menstrual cycle experienced by women who ran compared with women who did not regularly practice sports. A question was also asked about the runners' knowledge of FAT.

## MATERIALS & METHODS

### Participants

The study involved 217 women who regularly run (Group R), 32.1 ± 8.4 years, and 143 women in a control group (Group C) 31.9 ± 8.8 years. The women in Group R had been running on average (median) for 4 years. The minimum distances covered by the runners were, on average, 5 km, and the maximum 15 km. The women declared that their runs usually lasted an hour, and that they ran a minimum of three and a maximum of four times per week. All of the running, both short and long distances, was done outdoors. The vast majority of women from group C, about 90%, declared that they did not practice any sports activity on a regular basis. The remaining women in Group C only exercised rarely and irregularly. Their total weekly training time did not exceed 1.5 h. A summary of these attributes is presented in Table 1. Most of the respondents from both groups lived in Kraków and its vicinity, in the Małopolska Voivodeship (Lesser Poland voivodeship). Some of the women also came from the neighboring Śląsk Voivodeship (Upper Silesian voivodeship). They were mainly students from the university as well as their family and friends.

In both groups, the criteria for exclusion from the study was the use of hormonal contraception, or the declaration of amenorrhea caused by menopause, or another known factor not directly related to physical exercise such as hysterectomy, pregnancy, or polycystic

**Table 1** Anthropometric parameters, length of menstrual bleeding, number of regular monthly cycles, training characteristics, hydration and 'special diet' in groups R—runners ($N = 217$) and C—control ($N = 143$), (the Mann–Whitney $U$ test, the $\chi^2$ test, $p$—level of significance).

| Dependent variable | Average | | Medians | | Standard deviation | | $U$ | $p$ |
|---|---|---|---|---|---|---|---|---|
| | R | C | R | C | R | C | | |
| Age [years] | 32.06 | 31.90 | 32.00 | 31.00 | 8.38 | 8.85 | −.37 | .711 |
| Height [cm] | 166.24 | 167.17 | 165.00 | 168.00 | 5.92 | 6.35 | −1.49 | .135 |
| Body weight [kg] | 61.72 | 67.96 | 60.00 | 65.00 | 10.33 | 13.43 | −4.45 | <.001 |
| BMI [kg/m$^2$] | 22.30 | 24.28 | 21.63 | 23.34 | 3.29 | 4.40 | −4.44 | <.001 |
| How many days does menstruation last? | 4.79 | 5.27 | 5.00 | 5.00 | 1.25 | 1.11 | −3.67 | <.001 |
| How many regular menstrual cycles have you had in the last year? | 9.62 | 11.22 | 11.00 | 12.00 | 3.27 | 1.42 | −4.49 | <.001 |
| How many years have you been running? | 5.47 | – | 4.00 | – | 5.01 | – | – | – |
| What is the minimum distance you run? [km] | 6.88 | – | 5.00 | – | 3.70 | – | – | – |
| What is the maximum distance you run? [km] | 19.20 | – | 15.00 | – | 11.43 | – | – | – |
| What is the minimum number of times you run in a week? | 3.24 | – | 3.00 | – | 1.20 | – | – | – |
| What is the maximum number of times you run in a week? | 4.04 | – | 4.00 | – | 1.25 | – | – | – |
| What is the minimum number of hours for one run? | 1.03 | – | 1.00 | – | .38 | – | – | – |
| What is the maximum number of hours for one run? | 1.45 | – | 1.00 | – | .67 | – | – | – |
| What is the minimum number of litres of water you drink in a day? | 1.77 | 1.51 | 2.00 | 1.50 | .61 | .56 | −3.95 | <.001 |
| What is the maximum number of litres of water you drink in a day? | 2.43 | 1.66 | 2.50 | 1.50 | .76 | .64 | −6.27 | <.001 |

| | | Runners | | Control | | $\chi^2$ | $p$ |
|---|---|---|---|---|---|---|---|
| | | $N$ | % | $N$ | % | | |
| Do you use a 'special diet'? | Yes | 62 | 28.70 | 19 | 13.29 | 11.71 | 0.001 |
| Typ of 'special diet' of runners: | | $N$ | % | | | | |
| Vegan or vegetarian | | 25 | 11.52 | | | | |
| Balanced diet | | 10 | 4.61 | | | | |
| Low calorie | | 8 | 3.69 | | | | |
| High-protein | | 4 | 1.84 | | | | |
| Ketogenic | | 2 | 0.92 | | | | |
| Gluten free | | 3 | 1.38 | | | | |
| Other specific diets, such as diabetic and dairy-free diets | | 3 | 1.38 | | | | |
| Lak of information about type of 'special diet' | | 8 | 3.69 | | | | |

ovary syndrome. The lack of a correctly completed questionnaire also resulted in exclusion from the research.

Participation in the research was voluntary and anonymous in accordance with the Declaration of Helsinki and the participants were informed about the purpose of the research and their right to refuse to answer the survey questions. The research protocol was reviewed and approved by the Bioethical Committee of the Andrzej Frycz Modrzewski Krakow University (Permission number KBKA/93/O/2020). The study utilized a survey conducted on-line using Google Forms. Hence, the participants were anonymous to the authors of the study and vice versa. There was no way of receiving informed consent from

the participants. The questionnaire was completed by volunteers who wanted to share their own experience about menstrual disorders.

## Questionnaire

The questionnaire utilized in this study was composed by the authors. This was an unvalidated questionnaire that was used for the measurement of cycle changes. The original and English language translation of the questionnaire administered to study participants is included as Supplemental Information. The research was conducted on-line, primarily among the members of the university's sports club and their network of contacts. Additionally, the research used internet portals intended for people who were interested in running and linked with groups who participate in this type of sports activity. The control group survey was conducted on social messaging platforms intended for students attending the university where the study was conducted, as well as the female family and friends of these students. The survey contained questions on, among other things, details related to training sessions, including the duration of a single training session, the frequency of the training sessions, and the distance run in each training session. In addition, the questions in the survey concerned details about the monthly cycle of the women runners, as well as their type of diet, and the amount of water consumed daily. At the end of the survey, the runners were asked a question to verify their knowledge of the female athlete triad. The question was: "Do you know what Female Athlete Triad is? If YES, please write…"

The control group questionnaire was slightly modified. The detailed running questions were removed, as was the question "Do you use a 'special diet' for athletes, *e.g.*, low calorie, high protein, vegetarian?" This question was amended to one in which participants were asked to answer yes or no to the question "Do you use a 'special diet', *e.g.*, low calorie, high protein, vegetarian?" This question was followed by "If yes, which diet"?

In the research, the term 'special diet' was used to mean any diet used in a conscious manner or according to pre-established rules to which the respondents adhered, as well as any that the athletes themselves considered appropriate for them.

## Statistical analyses

Statistical analyses were performed using the SPSS 27 software program (Version 27.0, IBM Corp., Armonk, NY, USA). Two types of analyses were performed. The first compared the results of the group of running women and the control group. The second examined possible correlates and predictors of menstrual disorders.

Means, medians, standard deviations, as well as minima and maxima were used to describe numerical data (*continuous*) and for dichotomous and qualitative data - numbers and percentages. Due to the lack of normal distribution, the comparison of both groups for numerical variables was carried out using the Mann–Whitney $U$ test, and for dichotomous and qualitative variables the chi-square test was used. Correlations were calculated using Spearman's rho correlation coefficients. Multiple ordinal PLUM (Polytomous Logit Universal Model) regression analyses were used to analyse the predictors of menstrual disorders determined on the basis of the question "Have you ever missed your period for a long time after a period of regular bleeding? 1—Yes, for less than 3 months, 2—Yes, for

between 3 and 6 months, 3—Yes, for more than 6 months, 0—I have never had such a situation". The potential predictors of menstrual disorders that were investigated included age, body mass index (BMI), diet, hydration of the runners, and details of run intensity including their time, frequency, or distance. Results where $p < 0.05$ were considered statistically significant.

## RESULTS

Compared to the women who did not regularly participate in physical activity, the female runners were characterized by their lower body weight. Runners *vs.* control showed a median body weight 60 kg *vs.* 65 kg ($p < 0.001$) and median BMI 21.63 kg/m$^2$ compared to 23.34 kg/m$^2$ for that of controls ($p < 0.001$) (Table 1).

This research showed differences between the monthly cycle of women who run and women who lead a sedentary lifestyle. The female runners had shorter periods of monthly bleeding, with a mean of 4.79 days relative to 5.27 days for controls ($p < 0.001$), and the number of regular cycles over the year was smaller than in the control group with a mean of 9.62 cycles per year compared to 11.22 for that of the control group ($p < 0.001$) (Table 1). There was also a difference between runners and control related to the frequency with which they experienced 24-day menstrual cycles, with 10.14% of runners and 3.5% of controls experiencing cycles of this duration ($p < 0.05$). In addition, runners experienced menstrual cycles of 25–31 days less frequently than the control group: 75.58% of runners reported experiencing longer cycles compared to 86.71% of controls ($p < 0.05$) (Table 2). There were no statistically significant difference between the runner and control groups in the proportion of monthly cycles longer than 31 days (Table 2). Despite the difference in the length of monthly cycles described above, the intra-group analysis showed that both runners and the control group predominantly had cycles of 25–31 days. The incidence of this cycle duration was significantly more frequent than longer cycles (>31 days) or shorter cycles (<25 days). There was no statistical difference between the two groups of women when comparing how often they missed their menstrual periods or in the incidence of spotting between periods.

Female runners showed an increased incidence of painless menstrual bleeding relative to control (23.96% *vs.* 7.69%, respectively, $p < 0.05$). The female runners also reported fewer instances of menstruation that was painful only at the beginning of the bleeding period (70.97% *vs.* 84.62%, respectively, $p < 0.05$). The incidence of menstrual pain throughout the bleeding period did not differ significantly between the groups (Table 2). When reported on a numerical scale, the degree of pain did not differ significantly between groups. In the runners group, when asked "*whether, in their opinion, running leads to changes in the level of pain during menstruation*" 30.41% replied that running reduces pain, and only 0.92% said that running during menstruation increased pain. The dominant answer was that running did not modify the perceived degree of pain during menstruation (56.68%).

The runners, more often than the control group, used 'special diets' (28.70%) *vs.* (13.29%) (Table 2), and also drank more water. The most popular diet among the runners was a vegan or vegetarian diet, which was observed by 25 people (11.52%). A healthy

**Table 2 Characteristics of the monthly cycle of the women surveyed and the use of 'special diet' in the groups R—runners ($N = 217$) and C—control ($N = 143$) ($\chi^2$ test, $p$-level of significance for whole categorical variable).**

| | | Runners | | Control | | $\chi^2$ | $p$ |
|---|---|---|---|---|---|---|---|
| | | N | % | N | % | | |
| Age when menarche occurred | 9 | 11 | 5.07 | 2 | 1.40 | 11.49 | .321 |
| | 10 | 6 | 2.76 | 7 | 4.90 | | |
| | 11 | 20 | 9.22 | 15 | 10.49 | | |
| | 12 | 41 | 18.89 | 31 | 21.68 | | |
| | 13 | 64 | 29.49 | 46 | 32.17 | | |
| | 14 | 35 | 16.13 | 23 | 16.08 | | |
| | 15 | 23 | 10.60 | 12 | 8.39 | | |
| | 16 | 13 | 5.99 | 3 | 2.10 | | |
| | 17 | 1 | 0.46 | 2 | 1.40 | | |
| | 18 | 2 | 0.92 | 0 | .00 | | |
| | 19 | 1 | 0.46 | 2 | 1.40 | | |
| Have you experienced any loss of your monthly cycle? | Never | 117 | 53.92 | 94 | 65.73 | 5.30 | .151 |
| | Less than 3 months | 67 | 30.88 | 31 | 21.68 | | |
| | Between 3 and 6 months | 22 | 10.14 | 11 | 7.69 | | |
| | Over 6 months | 11 | 5.07 | 7 | 4.90 | | |
| How long is your monthly cycle? | Every 24 days | 22 | 10.14[a; x] | 5 | 3.50[b; x] | 7.80 | .020 |
| | 25–31 days | 164 | 75.58[a; y] | 124 | 86.71[b; y] | | |
| | More than 31 days | 31 | 14.29[a; x] | 14 | 9.79[a; x] | | |
| Is menstruation painful? | Painless | 52 | 23.96[a; x] | 11 | 7.69[b; x] | 16.11 | <.001 |
| | Painful at the beginning | 154 | 70.97[a; y] | 121 | 84.62[b; y] | | |
| | Painful throughout | 11 | 5.07[a; x] | 11 | 7.69[a,x] | | |
| The degree of pain experienced during menstruation | 0 | 15 | 6.91 | 10 | 6.99 | 4.98 | .893 |
| | 1 | 20 | 9.22 | 11 | 7.69 | | |
| | 2 | 18 | 8.29 | 7 | 4.90 | | |
| | 3 | 24 | 11.06 | 14 | 9.79 | | |
| | 4 | 20 | 9.22 | 16 | 11.19 | | |
| | 5 | 20 | 9.22 | 14 | 9.79 | | |
| | 6 | 36 | 16.59 | 20 | 13.99 | | |
| | 7 | 24 | 11.06 | 18 | 12.59 | | |
| | 8 | 24 | 11.06 | 24 | 16.78 | | |
| | 9 | 7 | 3.23 | 4 | 2.80 | | |
| | 10 | 9 | 4.15 | 5 | 3.50 | | |
| The effect of training on menstruation pain | Lessens the pain | 66 | 30.41 | – | – | – | – |
| | Increases the pain | 2 | 0.92 | – | – | – | – |
| | Has no effect | 123 | 56.68 | – | – | – | – |
| | I don't train during my period | 26 | 11.98 | – | – | – | – |

**Table 2** (*continued*)

| | | Runners | | Control | | $\chi^2$ | $p$ |
|---|---|---|---|---|---|---|---|
| | | **N** | **%** | **N** | **%** | | |
| Do you experience spotting between periods? | Yes | 54 | 25.00 | 33 | 23.08 | 0.17 | .677 |
| Do you use a 'special diet'? | Yes | 62 | 28.70 | 19 | 13.29 | 11.71 | .001 |

**Notes.**
Different letters (a, b) denote significant group differences for a given category of the categorical variable at $p < 0.05$, the same letters (a, a) denote lack of group differences for a given category of the categorical variable at $p < 0.05$. Different letters (x, y) denote differences between categories within groups at $p < 0.05$, the same letters (x, x) denote lack of differences between categories within groups at $p < 0.05$.

balanced diet was practiced by 10 people (4.61%), and a low-calorie diet was practiced by 8 people (3.69%). High-protein diets, ketogenic diets, and gluten-free diets were also reported by 4, 2, and 3 participants, respectively. Other specific diets, such as diabetic and dairy-free diets, were mentioned by 3 people (1.38%) and 8 people (3.69%) did not detail the exact diet. Runners drank a minimum of 2.00 L of water a day *vs.* only 1.5 L that was consumed by the control group (median, $p < 0.001$). Runners drank a maximum of 2.50 L relative to the 1.5 L consumed by control (median, $p < 0.001$) (Table 1).

For runners, correlation analysis showed that the older the women were, the shorter their monthly bleeding lasted, and it was less painful ($p < 0.01$) (Table 3). Less pain during menstruation was also experienced by runners who developed menarche at an older age, the athletes who reported running longer distances, and women who had longer training times per session. In the control group, the data showed the strongest correlation between increased age and a decrease in the perception of menstrual pain (Table 4).

Regression analyses were only performed for the runners (Fig. 1). The assumption of equal regression coefficients important for ordinal regression, for each predictor in each cumulative category of the dependent variable was satisfied $\chi^2(24) = 4.70$; $p = 0.999$. Nagelkerke's pseudo-R-square value was 0.25. The overall model fit test did not yield a statistically significant result $\chi^2(12) = 14.99$; $p = 0.242$, but some predictors were statistically significant (Fig. 1). The use of a 'special diet' was a positive predictor of amenorrhea (Fig. 1).

The runners were asked "*Do you know what Female Athlete Triad is? Please tell us what you think it is.* " In response to this question, 75.93% replied that they knew nothing about it. The remaining women replied with varying degrees of accuracy, mainly emphasizing that the triad was "amenorrhea" (15.74%), or that it was, for example, energy imbalance or eating disorders (2.77%). The triad syndrome, consisting of three interrelated elements, *i.e.,* eating disorder, menstrual disorders, and osteoporosis, was correctly described by only 5.56% of the women surveyed.

## DISCUSSION

Sports-related menstrual disorders are complex, multi-factorial conditions that can have serious consequences for the health of women. There are many factors that may contribute to menstrual disorders in athletics, including strenuous training, stress related to exercise and competition, body fat percentage, and genetic factors. It is, however, low energy availability which is the most likely predisposing cause (*Mountjoy et al., 2018*).

**Table 3   Analysis of the correlation between the features of the monthly cycle and anthropometric parameters, training characteristics and nutrition in the group of runners ($N = 217$).**

| Runners group | Has your monthly cycle disappeared? | How long does your monthly cycle last? | For how many days does bleeding take place | How many regular cycles have you had in the last year? | Is your period painful? | The degree of pain experienced during your period | Do you experience spotting between periods? |
|---|---|---|---|---|---|---|---|
| Age [years] | −.16[*] | −.03 | −.25[**] | .06 | −.22[**] | −.30[**] | .04 |
| Height [cm] | −.05 | −.16[*] | −.10 | .02 | −.07 | −.11 | −.02 |
| Body weight [kg] | −.03 | −.13 | −.11 | −.04 | .01 | −.02 | −.01 |
| BMI [kg/m²] | −.01 | −.08 | −.09 | −.05 | .03 | .02 | .01 |
| How many years have you been running? | .01 | −.09 | .13 | .06 | .04 | .02 | .06 |
| What is the minimum distance you run? [km] | −.01 | .06 | .00 | .05 | −.13 | −.13 | .06 |
| What is the maximum distance you run? [km] | .04 | .08 | .00 | −.04 | −.18[**] | −.08 | .02 |
| What is the minimum number of times you run in a week? | .11 | .01 | .01 | .02 | −.08 | .01 | .00 |
| What is the maximum number of times you run in a week? | .15[*] | .07 | .16[*] | −.07 | −.03 | .06 | .06 |
| What is the minimum number of hours for one run? | .12 | −.07 | −.06 | .06 | −.08 | .04 | .14[*] |
| What is the maximum number of hours for one run? | .00 | −.04 | −.13 | .00 | −.23[**] | −.12 | −.04 |
| Age when menarche occurred | −.04 | −.11 | −.03 | .04 | −.11 | −.21[**] | −.07 |
| Do you use a 'special diet'? | .11 | −.09 | −.09 | −.05 | −.07 | −.05 | −.01 |
| What is the minimum number of litres you drink in a day? | .12 | −.04 | −.05 | .08 | −.06 | −.14[*] | −.06 |
| What is the maximum number of litres you drink in a day? | .17 | −.04 | .01 | −.12 | .00 | −.11 | −.31[*] |

**Notes.**

Spearman's rho correlation [*]$p < 0.05$, [**]$p < 0.01$.

This study showed that the recreational runners, compared to women in the control group who did not regularly participate in physical activity, had a lower body weight and lower BMI. In addition, they had fewer regular monthly cycles per year, their menstrual bleeding was shorter, and they had slightly more spotting between menstrual periods. Moreover, it was found that the runners had cycles every 24 days, which was more often than the control group and cycles of 25–31 days less frequently than the control group. There was no difference between the groups in the proportion of cycles longer than 31 days, indicating that the runners did not suffer from infrequent menstruation. However, the question arises as to whether the increased frequency of short cycles in the runners might not be the first symptom in the shortening of the luteal phase and a prelude to the cessation of menstruation. However, despite the fact that 75.58% of the runners declared cycles ranging from 21 to 31 days, *i.e.*, falling within the definition of a regular cycle covering from 21 to 31 days, in our own research, the occurrence of ovulation in the examined women was not
**Table 4  Analysis of the correlation between the features of the menstrual cycle and anthropometric parameters as well as nutrition in the control group (N = 143).**

| Control group | Has your monthly cycle disappeared? | How long does your monthly cycle last? | For how many days does bleeding take place | How many regular cycles have you had in the last year? | Is your period painful? | The degree of pain experienced during your period | Do you experience spotting between periods? |
|---|---|---|---|---|---|---|---|
| Age [years] | −.16 | .04 | −.20[*] | .12 | −.16 | −.34[**] | −.08 |
| Height [cm] | −.08 | .01 | .03 | −.02 | .02 | .06 | .02 |
| Body weight [kg] | .14 | .04 | −.12 | −.11 | .01 | −.04 | .06 |
| BMI [kg/m2] | .19[*] | .02 | −.16 | −.11 | .00 | −.07 | .06 |
| At what age did menstruation first start? | .08 | .05 | .04 | .01 | .14 | .01 | −.02 |
| Are you on a 'special diet'? | .15 | −.07 | .19[*] | .01 | −.11 | −.12 | .03 |
| What is the minimum number of litres you drink in a day? | .17[*] | −.07 | .07 | −.15 | .07 | .02 | .01 |
| What is the maximum number of litres you drink in a day? | .11 | −.13 | .08 | −.10 | .04 | −.02 | .08 |

**Notes.**

Spearman's rho correlation *$p < 0.05$, **$p < 0.01$.

checked, and this is important because the research of *Shangold et al. (1990)* showed that, in spite of the occurrence of a regular monthly cycle, there are ovulation disorders and non-ovulatory cycles in those women who participate in sports. The authors found that even if sportswomen had a regular, standard monthly cycle, as many as 50% of the female runners with eumenorrhea were not ovulating compared with 83% of the control group. *De Souza (2003)* stated that disorders of the luteal phase, characterized by poor endometrial maturation as a result of inadequate progesterone production and short luteal phases, are associated with infertility and habitual spontaneous abortions. In recreational athletes, the 3-month sample prevalence and incidence rate of lymphoproliferative disorders and anovulatory menstrual cycles is 48% and 79%, respectively (*De Souza, 2003*). The above-mentioned studies, therefore, support the assertion that female athletes who have regular monthly cycles can, nevertheless, have problems such as chronic anovulatory cycles or a defect luteal phase (*Vilšinskaite et al., 2019*).

The incidence of secondary amenorrhea in the general population who do not participate in sports ranges from 2% to 5%. Reports on the frequency of amenorrhea in the sports population vary widely, mainly due to the differences in methodology between studies, and range from 6% to 79% (*Matzkin, Curry & Whitlock, 2015*; *De Souza et al., 2017*; *Beals & Meyer, 2017*; *Daily & Stumbo, 2018*; *Williams, Statuta & Austion, 2017*). This study showed that 30.88% of runners suffered from menstrual disorders lasting less than 3 months, 10.14% of runners between 3 and 6 months, and 5.07% of runners over 6 months. In the physically inactive control group, the percentage of women with similar disorders was 21.68%, 7.69% and 4.90%, respectively. The percentage of women in both groups who had

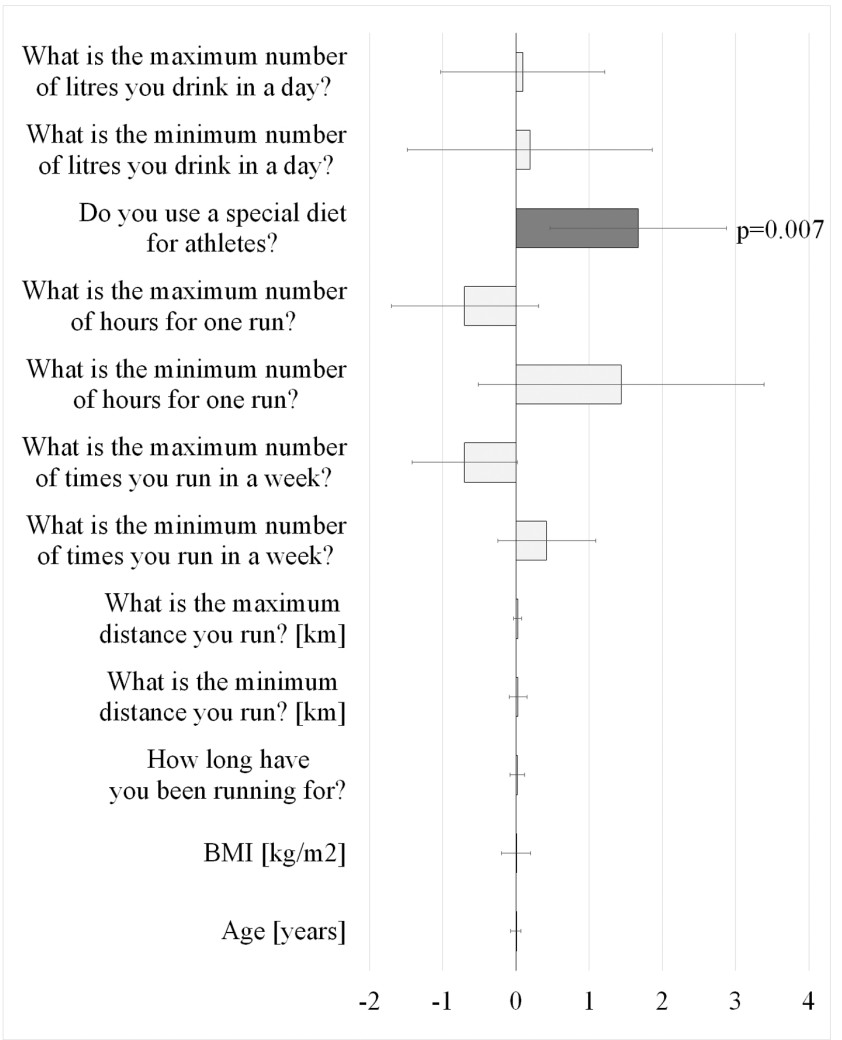

**Figure 1** Results of the PLUM coefficient regression of the predictors of amenorrhea in the group of female runners ($N = 217$), (Regression Coefficient and 95% confidence intervals).

never experienced a cessation of their menstrual cycle is more favourable for women in the control group, 65.73% *vs.* 53.92%. However, these differences did not reach statistical significance.

In both groups, significantly more women indicated that they felt pain only at the beginning of menstruation, rather than pain throughout the duration of menstruation, or painless menstruation. It was noteworthy that the runners reported much more often than the control group that their menstrual bleeding was painless. The differences between the groups in the assessment of pain on the numerical scale were not statistically significant. However, 30.41% of the women reported pain reduction as a result of running, indicating a possible analgesic effect of physical activity. These results may point to the analgesic and muscle-relaxing effect of movement, which is confirmed by other authors of studies assessing the impact of physical activity on symptoms related to the monthly cycle

(*Sammon, Nazareth & Petersen 2016*; *Vishnupriva & Rajarajeswaram, 2011*; *Dehnavi, Jafarnejad & Goghary, 2018*; *Azhary et al., 2005*).

Interestingly, the findings of the authors' own research has shown that the distance and time spent on each run were important in the assessment of the pain occurring during menstrual bleeding. Women running longer distances and spending more time on each run reported less pain during menstruation. There are several hypotheses in the literature on the subject of the effect of exercise on the relief of menstrual pain symptoms. One such hypothesis is that a diminution of menstrual symptoms occurs when the equilibrium is disturbed in the hypothalamic-pituitary-ovarian axis. This may be caused by exercise-induced stress or weight loss.

In addition, physical effort increases blood flow in the human body, improves metabolism, oxygenates tissues, and thus induces muscle relaxation. Furthermore, physical activity contributes to the release of beta-endorphins, which have an analgesic effect, and may therefore contribute to the alleviation of menstrual pain. It seems that the results of this research can be related to the above-mentioned hypotheses, and the longer the physical effort lasts, the longer the positive physiological effects associated with it are reflected in a woman's body (*Golomb, Solidum & Warren, 1998*).

In this study, the correlation analysis showed that the age of the runners correlated negatively with the cessation of the menstrual cycle, the duration of menstruation, the reported painfulness of the period, and the degree of pain during menstruation. Following correlation analysis, results in the runner group are similar to those obtained by other researchers of the subject, in which it has been shown that young athletes experience menstrual disorders much more often than mature women practicing the same sport, 67% *vs.* 9% (*De Souza et al., 2017*).

Less pain was also associated with runners who experienced menarche at an older age. The gynecologic age of the athlete seems to be even more important, *i.e.,* the difference between the chronological age and the menarche age. *Loucks (2003)* found in their research that, due to the lack of energy in patients with a gynecologic age from 14 to 18 years of age, the decrease in the frequency of luteinizing hormone secretion was not as high as in patients with a gynecologic age from 5 to 8 years. It has also been shown that women between the ages of 25 and 40 who participated in an exercise program, together with a calorie restricted eating plan, experienced only a few disruptions to the regularity of their menstrual cycle.

A woman's age is an important factor that has a significant impact on the course of the menstrual cycle. In this study, it was shown for both groups that the older a woman is, the lower the level of pain experienced during menstruation. This is reflected in the literature, where it has been shown that dysmenorrhea is more common in young women. For example, in the age range 17–24 years, dysmenorrhea affects 67% to 90% of women, while in adult women rates vary from 15% to 75%, and in mature women only 7% to 15% experience severe pain that limits their daily functioning (*Hong, Jones & Mishra, 2014*).

It would be outside the scope of this study to analyse the diet of athletes. Nevertheless, having the knowledge that menstrual disorders in sport occur mostly as a result of low energy availability resulting from dietary restrictions, the runners were asked whether
they followed any 'special diet' (*e.g.*, ketogenic, vegetarian, vegan) that was especially any intended for athletes. The nutritional habits of the runners in terms of the amount of nutrients contained in their meals were not studied. The question was aimed only at the general recognition of whether the women undertaking recreational running pay attention to what they eat, whether they have knowledge about the existence of diets for athletes, or whether they generally care about their eating habits used in sport.

The use of a 'special diet', *i.e.*, one that the athletes themselves considered appropriate, was a positive predictor of the occurrence of amenorrhea and the use of a 'special diet' was conducive to the cessation of the menstrual cycle. Among the runners, 28.70% declared that they used a 'special diet', while among the control group this was only 13.29%. When analysing the answers to the question about the types of diet, however, it is difficult to designate them as specific diets for athletes. A detailed analysis of the responses from runners showed that the most frequently declared diets were a vegan or vegetarian diet and, less frequently, a balanced/healthy diet. The authors of this study, however, are not in a position to confirm that the diet actually was well balanced and what, exactly, the runners considered a 'healthy' diet. Additionally, some of the women declared using a low-calorie diet, and some participants also referenced high-protein and gluten-free diets. Only two women reported using a specific diet for athletes. Such answers suggest that the runners lack knowledge about the obligation to supplement the caloric deficit, which may arise in connection with the practiced sports discipline, and could directly lead to disorders of the menstrual cycle. The results obtained indicate an urgent need to restore the energy balance in the bodies of the women runners. Following 'special diets' without dietician and nutritionist recommendations or guidance may contribute to females not ingesting enough to meet their energy needs.

The runners' knowledge about the FAT turned out to be negligible. In this study only 24.07% of the recreational runners declared they "*knew something*" about the concept of FAT in sport, which means that 75.92% did not know what syndrome they were being asked about. However, when analysing the accuracy of the answers given by the runners, it turned out that only 5.56% gave the correct answer, and most often the women stated that, according to them, only the loss of the monthly cycle was caused by athletic activity. It is disturbing that a lack of knowledge about FAT is common among people who have practiced sports on a regular basis for many years, albeit on a recreational basis (*Larsen et al., 2020*; *Kroshus, DeFreese & Kerr, 2018*).

Identifying FAT, especially in young female athletes, can sometimes be difficult as the health consequences may not be immediately apparent and therefore not easy to recognize. However, any woman with at least one component of FAT should be assessed for possible other components of the condition. Primary prevention should focus on education about the existence, signs, and symptoms of FAT in sport and this should be a priority among all health care professionals, coaches, and caregivers involved in the health and optimal psycho-physical condition of female athletes (*Klein, Paradise & Reeder, 2019*; *Lebrun, 2007*). Given the small amount known about FAT among healthcare representatives and trainers (*Berz & McCambridge, 2016*;

*Thomas, Erdman & Burke, 2016*), every effort should be made to include information about FAT in the curriculum of the various medical professions.

The limitations of the study included a lack of full dietary analysis, self-reported menstrual cycle characteristics, and determination of menstrual cycle features from survey *vs*. tracking for 3–6 months.

## CONCLUSIONS

Compared to women in the control group, who did not exercise, recreational runners had shorter and less regular monthly cycles and shorter monthly bleeding. It was also found that women running longer distances and spending more time on each run reported less pain during menstruation.

Based only on general information about the runners' diets, without any testing of the nutritional values contained in their meals, it can be concluded that type of diet may contribute to the cessation of the menstrual cycle.

### Funding
The authors received no funding for this work.

### Competing Interests
The authors declare there are no competing interests.

### Author Contributions
- Joanna Witkoś conceived and designed the experiments, performed the experiments, authored or reviewed drafts of the paper, and approved the final draft.
- Magdalena Hartman-Petrycka analyzed the data, prepared figures and/or tables, and approved the final draft.

### Human Ethics
The following information was supplied relating to ethical approvals (i.e., approving body and any reference numbers):

The study was conducted according to the guidelines of the Declaration of Helsinki, and approved by the Institutional Ethics Committee of the Andrzej Frycz Modrzewski Kraków University KBKA/93/O/2020, 23.01.2020.

### Data Availability
The raw data are available in the Supplemental Files.

### Supplemental Information
Supplemental information for this article can be found online at http://dx.doi.org/10.7717/peerj.12903#supplemental-information.

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
