# Peer review of "The Female Athlete Triad—the impact of running and type of diet on the regularity of the menstrual cycle assessed for recreational runners"

_PeerJ, doi:10.7717/peerj.12903_

## Round 0.1 · original submission · Major Revisions

Dear Authors,
Three experts in the field reviewed your present manuscript and retrieved several major issues you should consider before resubmitting the manuscript.

·

Basic reporting

Dear authors,
I appreciated the reading of your manuscript.
The treated topic appears interesting; however, the novelty in your findings is not clear with respect to the previous literature. Furthermore, the introduction and discussion section mainly focus on the role of physical activity on the menstrual cycle; however, the regression analysis shows diet as the only predictor of the menstrual cycle.
I have some major concerns, which I will explain further in the following comments, and I hope they will be useful in improving your manuscript.

TITLE
In my opinion, you investigated the differences between runners and non-runners rather than the impact of recreational running. Furthermore, only diet seems to be a predictor of menstrual cycles.

ABSTRACT
1. You did not investigate dietary patterns, but only the type of diet.
2. What is an “A proprietary survey?”
3. The menstrual cycle predictors appear different in the manuscript.

INTRODUCTION
1. The FAT abbreviation is not introduced.
2. Even though it is interesting, clear, and well written, the introduction is too long and explains details not investigated in the manuscript. The reader may lose the focus of the study and the real aim. I suggest the authors to shorten this part and introduce only the topic actually investigated in the research.

Experimental design

MATERIAL AND METHODS
1. PARTICIPANTS:
a. I suggest the authors be more specific about the amount and intensity of the runner group’s training. This could also be a factor to consider in the analysis as a covariate or a confounding factor. This is one of my major concerns, because women could train at different levels, intensities, frequencies, etc.
b. I also suggest the authors be more specific in describing the control group: was it completely inactive, did it not regularly practice some kind of sports or did it train occasionally?
c. Line 162 is a repetition of line 159.
2. QUESTIONNAIRE:
a. VAS should be explained.
b. In addition to the brief explanation of the questions, you should give some information about the possible answers.
c. Also in this case, I suggest using the type of diet or special diet and not dietary patterns, because you did not investigate dietary habits or the consumed food.
3. STATISTICAL ANALYSIS:
a. It is not clear if you used the SPSS version 21 or 27.
b. Lines 196-198 are superfluous.
c. Description of the normality assessment is not reported.
d. Why did you opt for an ordinal regression rather than a linear one?
e. You did not specify which variable for menstrual disorders has been used as a dependent variable and which variables have been considered predictors.

Validity of the findings

RESULTS:
1. Table 2: why it is reported gender differences in the caption?
2. I suggest commenting on all the significant or primary outcomes, because sometimes, in the discussion section, you explain some results without speaking about them in the result section.
3. The chi-square test is usually indicated with the symbol χ2.
4. In the regression analysis, you should report the results of the model (if the model is significant or not) and then the p of the single predictors inserted in the model.
5. Line 249: in table 5, only a special diet, not time without menstruation, seems to be a significant predictor.
6. Lines 250-255: this investigation is not reported in the study aim and the methods section.

DISCUSSION:
1. As previously said for the introduction section, I suggest the authors be more focused on their results and not excessively digress on topics not investigated through the questionnaires. Also in this case, the reader could lose the main findings and the novelty of your investigations.
2. In line with the previous comment, I suggest the authors reflect on the special diet as the only predictor of the menstrual cycle, while running has no effects. The authors should discuss this outcome.
3. Line 384: Female Athlete Triad is not abbreviated.

Reviewer 2 ·

Basic reporting

I think that the present paper has an important topic and a really well-intentioned endeavor that could help the literature to fill some of the gaps existing on this matter. I'm also positively impressed by the numerosity of the sample size.

However, a deep revision of the English language, which, I think. it is one of the main issues of this paper, is needed. Indeed, even if I overall got the meaning of most of the sentences, the syntax, structure, typos, and choice of vocabulary are extremely poor.

The material and methods section is also poorly described (i.e questionnaire section) not allowing the reader to understand much about the questionnaire and the participants. Additionally, why did you decide to use an invalidated questionnaire, you need to address this.

The discussion section should avoid reporting the results again. Also, I found that the flow of speech is not very linear and sometimes I had a hard time following.

Generally, all sections need to streamline and be more concise and clear.

Experimental design

look above

Validity of the findings

As above mention, the paper has a really good potential to be an excellent paper, however, the actual form is too primitive to be published.

Additional comments

I found quite irritating to start the reading of the paper and find a BIG TYPO on the first sentence such as FEMALE ATHLETE THRIAD!

·

Basic reporting

General comments:
I reviewed the article “An assessment of the impact that recreational running has on potential disturbances in the regularity of women’s menstrual cycles – a symptom of the Female Athlete Triad”. The article aims to evaluate the impact that recreational running has on menstrual cycle disorders comparing a group of runners with a group who did not regularly practice sports (control).
The present study indicated that the runners had shorter and less regular monthly cycles and shorter and more often painless menstrual bleeding comparing with control group. In addition, it seems that the frequency of menstrual disorders in runners was increased by following a special diet even if few details were reported about diet.
The manuscript is interesting and the topic has the potential to be interesting and useful from a scientific point of view but need to be improved in all its sections in order to understand better the research.

INTRODUCTION
I think that the introduction is too long especially referring to the endocrine and energy availability components. I suggest to summarize this part reporting only what is really important for your research.

Experimental design

MATERIALS AND METHODS
Physical activity:
Referring to the aim of the study, details of the recreational running are missing. We obtained a lot of general information from the questionnaire but we don’t know the features of each session of running. How many sessions are on short or long distance? What is the real length of one session? It is conducted outdoor or on the track?

Questionnaire:
Line 181: is better to specify that this survey is for runners and I think is not correct to report “intensity of training” because you don’t properly assess it.

Validity of the findings

RESULTS
I suggest to add another table to report all the information about water and special diets (lines:232-238) even if the authors have conducted two different analysis.
Line 210 you have to report also the BMI as a result.
Line 213: add table 1 and 2 in the brackets.
Line 223: more or less pain? In the table 2 you reported painless and here pain. Please verify the data.
Line 226: specify that the incidence of menstrual pain throughout the bleeding period was less frequent in the runners even if there are not difference between the groups.
Line 231: add table 2 in the brackets.
Lines 239-242: I suggest to report and discuss all the significant correlations and not only the age. Specifically, line 239 specify that the age is the main one factor influencing the monthly cycle in both groups but not the only one. Lines 240-241, the results about the amenorrhea and menstruation pain are not referring to both groups but only for runners.
Line 249: I suggest to remove the period “extended the length of time without menstruation”.

TABLES
Table 1: add the number of participants of each group and specify under the table the statistical analysis and the p value (<0.05). Correct the unit of measure of BMI.
Table 2: complete the caption. Specify under the table the statistical analysis and the p value (<0.05). What do you mean with gender differences? Specify what is “a or b”, “x or y” under the table.
Table 3: add the number of participants.
Table 4: add the number of participants and correct the unit of measure of BMI.
Table 5: correct the unit of measure of BMI.

DISCUSSION and CONCLUSIONS
As I suggest in the introduction, I think is better to focus the discussion only on your findings without digress on other aspects.
Lines 289-290: control your data because previously you say that cycles of 25-31 days are common in both groups.
Line 295: correct into “from 25 to 31 days”.
Line 333: add in the runner group after correlation analysis.
Lines 356-357: pay attention to say that the use of special diet was conducive to the disappearance of the menstrual cycle. Actually you don’t assess properly the nutritional habits of the runners or which diet has more or less effect, so I think is too strong as a conclusion.
Line 404: change “more often” with “not”. Also in the conclusions, I suggest to not be too assertive on the role of the diet.

---

## Round 0.2 · Minor Revisions

Dear Authors,

Three experts reviewed your manuscript and raised some minor points that you should consider before publication.

·

Basic reporting

The authors could consider to add a figure or substitute one of the tables with a figure.
it is not mandatory.

Experimental design

no comment

Validity of the findings

no comment

Additional comments

I suggest the authors insert only the useful abbreviation; for example, BMD or IOC, used only once, could not be abbreviated.
Furthermore, it would be best to be more consistent with the introduced abbreviations. Indeed, you used Group R and Group C only in the "Participants" section, while in the remaining text, you used runners and non-runners.

Reviewer 2 ·

Basic reporting

Congratulation for the big commitment and the exquisite work done in addressing all the comments, I do like the new version a lot.

Experimental design

okay

Validity of the findings

okay

·

Basic reporting

I have no additional comments to do.

Experimental design

I have no additional comments to do.

Validity of the findings

RESULTS
I think is a good idea to put the data referring water and special diets in table 1 even if the authors are not convinced.
In addition, pay attention because at the beginning of the results you report body weight with numerical data of BMI and not both the results (body weight and BMI).

Additional comments

I received the revised manuscript for review. I'm glad to see that the authors took reviewers' advice into consideration. The current manuscript is better than the one before it.

---

## Round 0.3 · accepted · Accept

Dear Authors,

The reviewers are fine with your revisions and think that the manuscript could be acceptable in the present form.

·

Basic reporting

No comments

Experimental design

No comments

Validity of the findings

No comments

Additional comments

No comments